# Magnetic and Transport Properties of New Dual-Phase High-Entropy Alloy FeRhIrPdPt

**DOI:** 10.3390/ma14112877

**Published:** 2021-05-27

**Authors:** Kohei Baba, Naoki Ishizu, Terukazu Nishizaki, Jiro Kitagawa

**Affiliations:** 1Department of Electrical Engineering, Faculty of Engineering, Fukuoka Institute of Technology, 3-30-1 Wajiro-Higashi, Higashi-ku, Fukuoka 811-0295, Japan; s1751036@bene.fit.ac.jp (K.B.); mem18102@bene.fit.ac.jp (N.I.); 2Department of Electrical Engineering, Faculty of Science and Engineering, Kyushu Sangyo University, 2-3-1 Matsukadai, Higashi-ku, Fukuoka 813-8503, Japan; terukazu@ip.kyusan-u.ac.jp

**Keywords:** high-entropy alloys, Fe, 4*d* and 5*d* transition metals, magnetic properties, dual-phase

## Abstract

High-entropy alloys (HEAs) are broadly explored from the perspective of mechanical, corrosion-resistance, catalytic, structural, superconducting, magnetic properties, and so on. In magnetic HEAs, 3*d* transition metals or rare-earth elements are well-studied compositional elements. We researched a magnetic HEA containing Fe combined with 4*d* and 5*d* transition metals, which has not been well investigated, and found a new dual-phase face-centered-cubic (fcc) HEA FeRhIrPdPt. The structural, magnetic, and transport properties were evaluated by assuming that FeRhIrPdPt is a mixture of FeRh_4_, FeIr_4_, FePd_4_, and FePt_4_, all with the fcc structure. The dual-phase is composed of a Rh- and Ir-rich main phase and a Pd- and Pt-rich minor one. FeRh_4_ and FeIr_4_ show spin freezings at low temperatures, while FePd_4_ and FePt_4_ are ferromagnetic. Two magnetic features can characterize FeRhIrPdPt. One is the canonical spin-glass transition at 90 K, and the other is a ferromagnetic correlation that appears below 300 K. The main and minor phases were responsible for the spin-glass transition and the ferromagnetic correlation below 300 K, respectively.

## 1. Introduction

High-entropy alloys (HEAs) were initially proposed for simple crystal structures such as face-centered cubic (fcc), body-centered cubic (bcc), and hexagonal close packing (hcp), in which more than five elements, each having an atomic fraction between 5% and 35%, randomly occupy one crystallographic site [1,2]. The HEA concept is now adopted in various crystal structures [3,4]. The high-entropy state contributes to the stability of the solid solution with the desired crystal structure through the relatively large mixing entropy. For example, (Gd_1/6_Tb_1/6_Dy_1/6_Tm_1/6_Yb_1/6_Lu_1/6_)_2_Si_2_O_7_ disilicate shows outstanding thermal stability, which is ascribed to the high-entropy state at the rare-earth site [5]. The significant atomic disorder leads to the severe lattice distortion effect. In particular, bcc HEAs show superior mechanical properties, attracting a great deal of interest. There is also growing interest in the study of HEAs with high ductility, refractory HEAs, those demonstrating superconductivity, as well as HEAs used in anti-corrosion coatings, biomaterials, catalysts, shape memory alloys, magnetic refrigeration materials, and so on [6,7,8,9,10,11,12,13,14,15].

In some HEAs containing 3*d* transition (Cr, Mn, Fe, Co, and Ni) or rare-earth metals, the magnetic properties are well investigated. Al_x_CoCrFeNi exhibits a structural change from fcc to bcc through the mixture of fcc and bcc as *x* is increased [16]. Al-free CoCrFeNi is ferromagnetic below 120 K, and the Curie temperature *T*_C_ tends to be enhanced above room temperature with increasing Al content. In some compositions, spin freezing appears at low temperatures, which is detected as the irreversible magnetization between zero-field-cooled (ZFC) and field-cooled (FC) states. The electrical resistivity of Al_x_CoCrFeNi shows metallic temperature dependence, whereas it is rather small due to atomic disorder. NiFeCoCrPd and NiFeCoCrMn Cantor–Wu alloys are famous equimolar HEAs. Both alloys are ferromagnets with *T*_C_ = 440 K for NiFeCoCrPd and *T*_C_ = 38 K for NiFeCoCrMn, respectively [17,18]. NiFeCoCrMn also shows a spin-glass behavior. The temperature dependences of the electrical resistivity of Cantor–Wu alloys are also weak but metallic. Although the almost-magnetic HEAs display ferromagnetism, Cr_20_Mn*_x_*Fe*_y_*Co_20_Ni*_z_* (*x* + *y* + *z* = 60) is reported to be an antiferromagnet with the transition temperature ranging from 80 to 190 K, depending on the composition [19]. As for rare-earth-based HEAs, Gd_20_Dy_20_Er_20_Ho_20_Tb_20_ forms a single-phase hcp structure and shows relatively good magnetocaloric properties [15].

Research into HEA magnetic materials also shows a growing interest in tailoring magnetic properties by changing the microstructure, especially in dual-phase HEAs as in the present study. In Fe_15_Co_15_Ni_20_Mn_20_Cu_30_, a spinodal decomposition occurs after heat treatment [20]. The spinodally decomposed HEA possesses enhanced *T*_C_ and magnetization by 48% and 70%, respectively, compared to the homogenized single-phase state. The control of magnetic properties was also achieved in dual-phase CoFeNi_0.5_Cr_0.5_-Al*_x_* (*x* = 0, 1.0, and 1.5) or multiphase Al_0.1_(Fe_1+*x*_CoCr_1−*x*_Mn)_0.9_ (*x* = 0, 0.2, 0.4, 0.6, and 0.8) HEAs [21,22]. The magnetic phases contained in the dual-phase HEAs independently exhibit their magnetic properties.

Up to now, HEAs containing Fe combined with 4*d* and 5*d* transition metals, such as Rh, Ir, Pd, and Pt, have not been well investigated. Here we show the results of the new equimolar FeRhIrPdPt. We found that this HEA forms a dual-phase fcc structure, which shows a spin-glass transition at approximately 90 K and a ferromagnetic correlation below 300 K. The phase assignment of these two characteristic magnetic properties was performed by employing FeRh_4_, FeIr_4_, FePd_4_, and FePt_4_ under the assumption that FeRhIrPdPt is a mixture of the four binary alloys with fcc structure. This article reports structural investigations of FeRhIrPdPt, FeRh_4_, FeIr_4_, FePd_4_, and FePt_4_. The magnetic and transport properties of these alloys were examined by measuring the temperature dependences of magnetization, isothermal magnetization curve, and electrical resistivity.

## 2. Materials and Methods

Polycrystalline samples, as listed in Table 1, were prepared by a homemade arc furnace using constituent elements Fe (99.9%), Rh (99.9%), Ir (99.99%), Pd (99.9%), and Pt (99.9%) under an Ar atmosphere. The elemental metals were arc-melted to button-shaped samples with a mass of 1.5 g and quenched on a water-chilled Cu hearth. The samples were flipped and remelted several times to ensure homogeneity. In this study, all samples received neither heat treatment nor deformation (e.g., rolling). Room-temperature X-ray diffraction (XRD) patterns of as-cast samples were collected using an X-ray diffractometer (XRD-7000L, Shimadzu, Kyoto, Japan) with Cu-Kα radiation in Bragg–Brentano geometry. We used thin slabs cut from the samples due to their high ductility.

The metallographic examination of the prepared sample was carried out by a field-emission scanning electron microscope (FE-SEM; JSM-7100F, JEOL, Akishima, Japan). The atomic composition in each sample area was investigated by an energy-dispersive X-ray (EDX) spectrometer equipped to the FE-SEM by averaging several data collection points. The elemental mappings of FeRhIrPdPt were also obtained by the EDX spectrometer.

The temperature dependence of dc magnetic susceptibility *χ*_dc_(*T*) between 50 K and 400 K and the isothermal magnetization curve were measured by VersaLab (Quantum Design, San Diego, CA, USA). The temperature dependence of ac magnetic susceptibility *χ*_ac_(*T*) between 70 K and 170 K under the ac field of 15 Oe and the frequency ranging from 20 Hz to 5 kHz was also measured using this apparatus. *χ*_dc_(*T*) below 100 K and the isothermal magnetization curve at 2 K were measured by MPMS3 (Quantum Design, San Diego, CA, USA). The high-temperature *χ*_dc_(*T*) between 400 K and 450 K was checked by a vibrating sample magnetometer (BHV-50H, Riken Denshi, Yutenji, Japan). The temperature dependence of electrical resistivity ρ (*T*) between 3 K and 300 K was measured by a dc four-probe method using a homemade system in a GM refrigerator (UW404, Ulvac cryogenics, Kyoto, Japan). In this equipment, we also measured *χ*_ac_(*T*) between 3 K and 300 K under the ac field of 5 Oe at 800 Hz.

## 3. Results and Discussion

Figure 1 shows the XRD patterns of prepared samples. In each pattern, the diffraction peak positions match those of the fcc structure with the lattice parameters listed in Table 1. The lattice parameters were determined by the unit cell parameter refinement program CellCalc [23]. In the binary alloys, the lattice parameter decreases from FePt_4_, FePd_4_, and FeIr_4_ to FeRh_4_. The lattice parameter of FeRhIrPdPt is almost intermediate between those of FePd_4_ and FeIr_4_. SEM images of the studied samples are displayed in Figure 2a–e. The images of all binary alloys demonstrate their single-phase nature (Figure 2a–d). In each alloy, the chemical composition determined by EDX analysis is consistent with the starting composition listed in Table 1. In contrast, the SEM image of FeRhIrPdPt shows the main phase with the bright image (70 vol.%, 74 wt.%), accompanying the minor phase (30 vol.%, 26 wt.%) precipitated between grain boundaries of the main phase (see the dark image in Figure 2e). Figure 2f shows the elemental mappings of FeRhIrPdPt, especially the Ir and Pd mappings that support the dual-phase nature. For further elucidation of the dual-phase nature in FeRhIrPdPt, transmission electron microscopy images and selected area electron diffraction patterns are desirable. As shown in Table 1, compared to the ideal composition Fe_20_Rh_20_Ir_20_Pd_20_Pt_20_, one can say that the main (minor) phase is Rh- and Ir-rich (Pd- and Pt-rich) alloy. We guess that the minor phase also forms an fcc structure with a lattice parameter close to that of the main phase because no extra XRD peak other than the fcc structure in Figure 1 was observed. We tried simulating the XRD pattern of FeRhIrPdPt by the superposition of the main and minor phases. The lattice parameter of each phase was estimated by the composition-weighted average of FeRh_4_, FeIr_4_, FePd_4_, and FePt_4_. The obtained parameters are 3.819 Å for the main phase and 3.845 Å for the minor one. The resultant XRD pattern calculated using PowderCell 2.3 software [24] with the weight fractions mentioned above is shown in Figure 1. The simulated pattern agrees with the experimental one.

Next, we discuss the static and dynamic magnetic properties within the framework of molecular field. The magnetic properties investigated by *χ*_dc_(*T*) and *M*–*H* (*M*: magnetization, *H*: external field) curve measurements are summarized in Figure 3a–d for FeRh_4_ and FeIr_4_. As shown in Figure 3a,c, the 1/*χ*_dc_(*T*) of FeRh_4_ and FeIr_4_ show linear temperature dependences (red lines), which are well-described by the Curie–Weiss law as follows:*χ*_dc_ (*T*) = *C*/(*T* − *Θ*_CW_),(1)
where *C* is the Curie constant and *Θ*_CW_ is the Weiss temperature. The effective magnetic moment *μ*_eff_ is derived from *C*. *μ*_eff_ and *Θ*_CW_ were obtained as listed in Table 2. In each alloy, the value of *μ*_eff_ per Fe atom is close to that of the Fe^3+^ ion (5.92*μ*_B_), which means a well-localized magnetic moment, and the negative *Θ*_CW_ indicates a dominance of antiferromagnetic (AFM) interaction between Fe atoms. The low-temperature *χ*_dc_(*T*) under ZFC and FC conditions with *H* = 100 Oe for FeRh_4_ is given in Figure 3b. Irreversibility between the temperature dependences of ZFC and FC magnetization is common in spin-glass materials. Additionally, the isothermal *M*–*H* curve at 2 K demonstrates a featureless magnetization process (see the inset of Figure 3b). FeIr_4_ also shows a spin-glass-like behavior (Figure 3d), whereas there are two anomalies, for example, at approximately 23 and 69 K in *χ*_dc_(*T*) under the FC state. The comparison of *M*–*H* curves in the inset of Figure 3d suggests no pronounced feature characteristic for ferromagnetic (FM) or AFM ordering.

FePd_4_ and FePt_4_ show FM transitions, which are observed as rapid increases of *χ*_dc_(*T*) below approximately 400 K and 200 K, respectively (Figure 4a,c). For FePt_4_, an additional transition is observed at 317 K, which can be confirmed by the small anomaly in the temperature derivative of *χ*_dc_ (see the inset of Figure 4c). The FM nature of these alloys was also checked by the *M*–*H* curves, as shown in the insets of Figure 4b,d. Although the *M*–*H* curves of FePd_4_ measured in increasing and decreasing *H* show no noticeable difference, a small hysteresis appears in FePt_4_. The 300 K isothermal *M*–*H* curve of FePt_4_ already displays an FM feature. Thus, the additional anomaly at 317 K can be ascribed to the entrance into the FM state. We note here that FeIr_4_, the alloy between Fe and 5*d* element, also possesses two magnetic anomalies. The *T*_C_ of each ferromagnet was determined as a minimum on the temperature derivative of *χ*_dc_(*T*) (see the inset of Figure 4a or Figure 4c). This procedure is frequently employed in ferromagnets [25,26,27]. The *χ*_dc_(*T*)s of FePd_4_ and FePt_4_ also show Curie–Weiss behaviors above their *T*_C_s (red lines in Figure 4b,d). The value of each *μ*_eff_ supports the localized magnetic moment for the Fe atom, and the positive *Θ*_CW_ is consistent with the FM ground state.

For FeRhIrPdPt, *χ*_dc_ was measured as a function of temperature under ZFC and FC protocols with an external magnetic field of 100 Oe (Figure 5a). Both *χ*_dc_(*T*)s show broad peaks at approximately 90 K, associated with the difference between ZFC and FC data, due to a spin-glass transition as mentioned below. Besides, the irreversible *χ*_dc_(*T*) between ZFC and FC states already appears below approximately 300 K, which suggests a growth of some magnetic correlation. The inverse *χ*_dc_(*T*) shows the straight line above 325 K and can be explained by the Curie–Weiss law with *μ*_eff_ = 3.40*μ*_B_/Fe and *Θ*_CW_ = 308 K (Figure 5b). The effective moment slightly smaller than those of binary alloys is probably due to the influence of the dual phase. The positive *Θ*_CW_ indicates the presence of FM correlation. Note that *Θ*_CW_ is comparable to 300 K, below which the irreversible *χ*_dc_(*T*) grows. So, the irreversibility of a magnetic domain could be responsible for the difference of *χ*_dc_(*T*) between ZFC and FC below 300 K. The FM correlation is further supported by the *M*–*H* curves depicted in Figure 5c,d. Although the isothermal magnetization at 350 K shows a paramagnetic behavior, a rather rapid increase of magnetization at the lower field appears below 300 K, close to *Θ*_CW_. With decreasing temperature, the rapid increase at the lower field is pronounced, whereas no saturation is confirmed, which indicates an additional contribution of paramagnetic moments. There seem to be two characteristic temperatures in the magnetism of FeRhIrPdPt: 90 K (spin-glass transition) and 300 K (FM correlation). Especially for the latter temperature, it may be important to investigate a possible long-range ordering. *χ*_ac_(*T*) is a useful tool for this because the real part of *χ*_ac_(*T*) can often distinguish respective magnetic ordering as a peak, even in the case that a large *χ*_dc_ anomaly due to a strong magnetic interaction masks a small anomaly by a minor magnetic ordering [28,29]. The inset of Figure 5a presents the real part of ac magnetization *χ*_ac_′(*T*) between 3 K and 300 K, showing no anomaly except for the broad peak due to the spin-glass transition. Therefore, the FM correlation below 300 K would not evolve into a long-range ordering. We note that a similar phenomenon was also observed in another magnetic compound, and is ascribed to an inhomogeneous magnetic state [30,31].

To assess the possible spin-glass transition in FeRhIrPdPt, we measured *χ*_ac_′(*T*) under several ac frequencies, as shown in Figure 6. *χ*_ac_′(*T*) shows a broad peak at *T*_f_ac_, and as anticipated for a spin-glass transition, *T*_f_ac_ slightly shifts to higher temperatures as the frequency is increased. We employed two models to explain the shift quantitatively [32]. One is the Vogel–Fulcher law, expressed by
*τ* = *τ*_0_*∙*exp [*E*_a_/(*k*_B_(*T*_f_ac_ − *T*_SG_))].(2)

The other model is the critical scaling approach, as follows:*τ* = *τ*_0_*∙*[*T*_f_ac_/*T*_SG_ − 1]^−*zv*^.(3)

In these equations, τ, τ_0_, *E*_a_, *T*_SG_, *zv*, and *k*_B_ are the inverse ac frequency, the spin relaxation time, the activation energy, the spin-glass temperature, the dynamical critical exponent, and the Boltzmann constant, respectively. Figure 7a is the ln *τ* vs. 1/(*T*_f_ac_ − *T*_SG_) plot by the Vogel–Fulcher law, which yields the parameters listed in Table 3. The relations *E*_a_ ~ 2*T*_f_ac_ and τ_0_ on the order of 10^−14^ s are consistent with those found in many canonical spin-glass systems [33,34]. The fitting result by the critical scaling approach is given in Figure 7b with a ln *τ* vs. ln(*T*_f_ac_/*T*_SG_ − 1) plot. The obtained parameters again fall in the typical range for spin-glass systems [32]. Furthermore, the Mydosh parameter *K* defined by Δ*T*_f_ac_/*T*_f_ac_log(Δ*f*) (where *f* is the ac frequency) is 0.007, comparable to those of canonical spin-glasses such as CuMn and AgMn [35]. Thus, FeRhIrPdPt can be regarded as a canonical spin-glass HEA.

Figure 8 shows the ρ (*T*) of investigated alloys. Each ρ is normalized by the room-temperature value listed in Table 2. The binary alloys show the metallic behavior; ρ decreases as the temperature is lowered. For the disordered FePd_4_, the temperature dependence is rather large in the ferromagnetic state, which is also observed in the other disordered ferromagnets [36]. Contrasted to the metallic behavior of binary alloys, ρ of FeRhIrPdPt slightly increases on cooling, which is a signature of carrier localization and may be due to the atomic disorder higher than those in the binary alloys. It should be noted that the other spin-glass HEAs show metallic temperature dependences [16,18,37].

Here we discuss the dual-phase formation in FeRhIrPdPt. Compared to the ideal equimolar Fe_20_Rh_20_Ir_20_Pd_20_Pt_20_, the EDX analysis indicates that the main phase is rich in Rh and Ir and poor in Fe, Pd, and Pt, whereas the minor phase shows the reverse tendency (see also Table 1). The enthalpy of formation Δ*H*_f_ of each binary compound for constituent elements [38] calculated by Troparevsky et al. can explain the dual-phase formation. Table 4 presents the Δ*H*_f_ of stable binary alloy for each component. Although Fe tends to form an alloy with all elements because all pairs Fe–Rh, Fe–Ir, Fe–Pd, and Fe–Pt show negative Δ*H*_f_, Fe–Pd and Fe–Pt pairings would be more robust. Furthermore, alloying between Ir and Rh (Pt and Pd) is also favored. These facts based on the pairwise Δ*H*_f_ can explain the dominant elements in each phase of FeRhIrPdPt. A similar discussion has been found in AuPdAgPtCu, also showing a dual-phase microstructure [39]. All binary alloys of noble metals (Rh, Ir, Pd, and Pt) with 1:1 composition form fcc [40,41]. Furthermore, many noble metal HEAs (e.g., PdPtRhIrCuNi and PdPtRhRuCe) are reported to be fcc alloys [39,42]. Therefore, the fcc FeRhIrPdPt would be deeply related to the existing phase diagrams of noble metals.

It is straightforward that the main and minor phases of FeRhIrPdPt independently exhibit their magnetism as in the other dual-phase HEA magnetic materials. The magnetic properties of FeRhIrPdPt can be characterized by the canonical spin-glass plus high-temperature FM correlation. So, in FeRhIrPdPt, the different phases would be responsible for the spin-glass transition and the high-temperature FM correlation. In the binary alloys, FeRh_4_ and FeIr_4_ are spin-glass materials, while FePd_4_ and FePt_4_ are ferromagnets. Based on the atomic ratios in the dual-phase of FeRhIrPdPt, we speculate that the main phase rich in Rh and Ir and minor one rich in Pd and Pt exhibit the spin-glass transition and the FM correlation, respectively. As mentioned in the Introduction, dual-phase HEAs would be useful in tailoring magnetic properties via a change of microstructure. In this case, the occurrence of spinodal decomposition is mainly responsible for the change of microstructure. Although the as-cast dual-phase HEA was investigated in this study, it is important to understand the thermal stability of the alloy in order to understand and control the microstructure. This is a future task. The other interesting aspect of dual-phase HEAs is a possible multifunctional magnetic material, in which two magnetic phases offer different magnetic functions.

## 4. Summary

We synthesized the new dual-phase fcc HEA FeRhIrPdPt and evaluated its structural, magnetic, and transport properties by assuming that the HEA is a mixture of FeRh_4_, FeIr_4_, FePd_4_, and FePt_4_, all with fcc structure. The dual-phase alloy is composed of a Rh- and Ir-rich main phase with 70 vol.% (74 wt.%) and a Pd- and Pt-rich minor one with 30 vol.% (26 wt.%). The minor phase could also form an fcc structure with a lattice parameter near that of the main phase. In the binary alloys, FeRh_4_ and FeIr_4_ are spin-glass materials dominated by antiferromagnetic interactions, whereas FePd_4_ and FePt_4_ are ferromagnetic. FeRhIrPdPt can be characterized by two magnetic features: one is the canonical spin-glass transition at 90 K, and the other is the ferromagnetic correlation revealed by the irreversible *χ*_dc_(*T*) under ZFC and FC conditions below 300 K. The latter correlation would not lead to long-range magnetic ordering. Based on each chemical composition of the dual-phase alloy, the main and minor phases would be responsible for the spin-glass transition and the ferromagnetic correlation at high temperatures, respectively. The temperature dependence of the electrical resistivity of FeRhIrPdPt indicates weak carrier localization, which is contrasted with the metallic behaviors of binary alloys. The dual-phase formation of FeRhIrPdPt can be discussed based on the pairwise formation enthalpies of the binary alloys.

## Figures and Tables

**Figure 1 materials-14-02877-f001:**
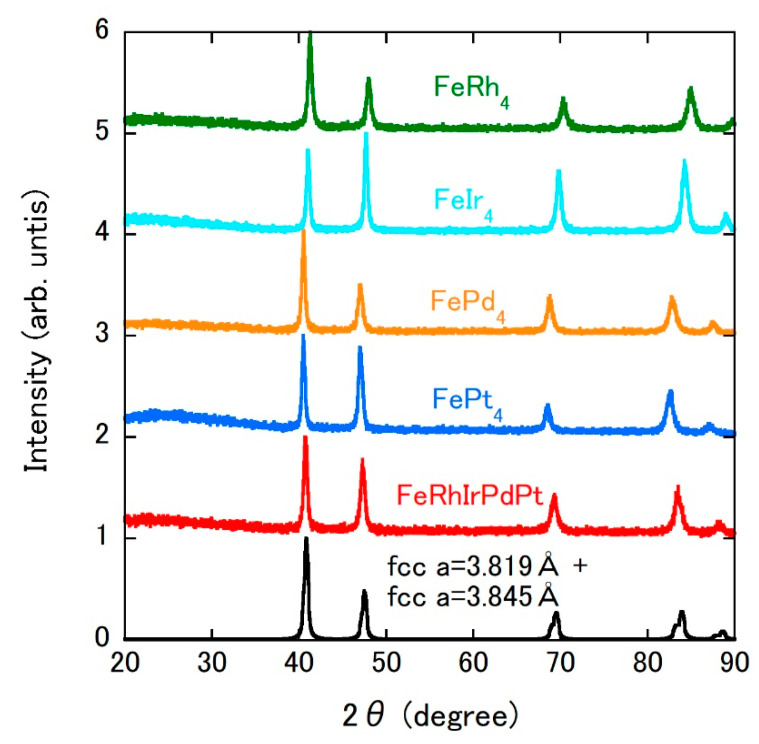
X-ray diffraction patterns of FeRh_4_, FeIr_4_, FePd_4_, FePt_4_, and FeRhIrPdPt. The simulated pattern for FeRhIrPdPt is also shown. The origin of each pattern is shifted by an integer value for clarity.

**Figure 2 materials-14-02877-f002:**
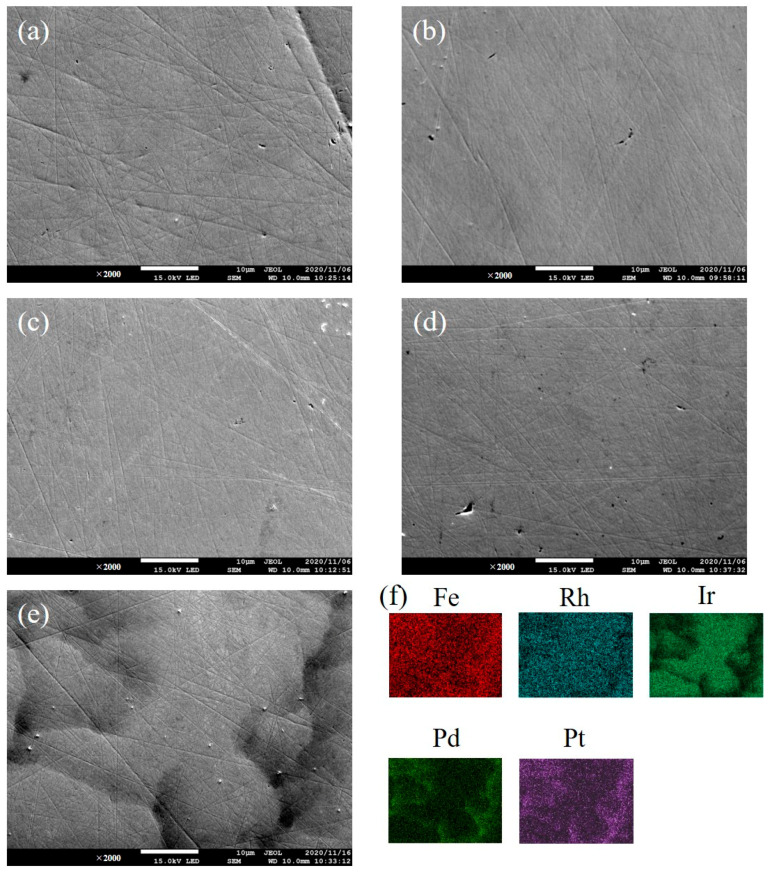
SEM images of (**a**) FeRh_4_, (**b**) FeIr_4_, (**c**) FePd_4_, (**d**) FePt_4_, and (**e**) FeRhIrPdPt, respectively. (**f**) Elemental mappings of FeRhIrPdPt.

**Figure 3 materials-14-02877-f003:**
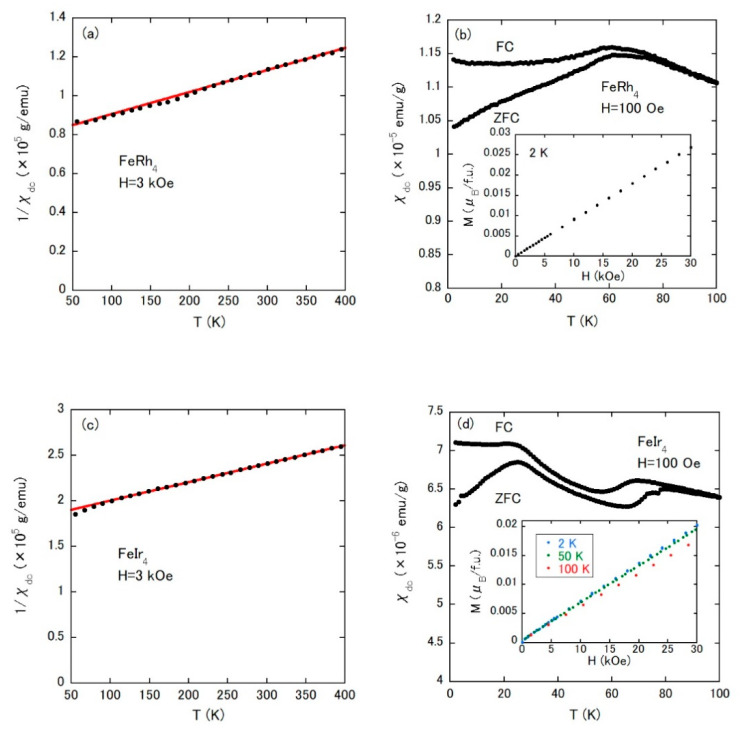
(**a**) Temperature dependence of 1/*χ*_dc_ under an external field of 3 kOe for FeRh_4_. (**b**) Temperature dependence of *χ*_dc_ under ZFC and FC conditions with an external field of 100 Oe for FeRh_4_. The inset is the isothermal *M*–*H* curve at low temperature. (**c**,**d**) Experimental results of FeIr_4_, corresponding to (**a**,**b**), respectively.

**Figure 4 materials-14-02877-f004:**
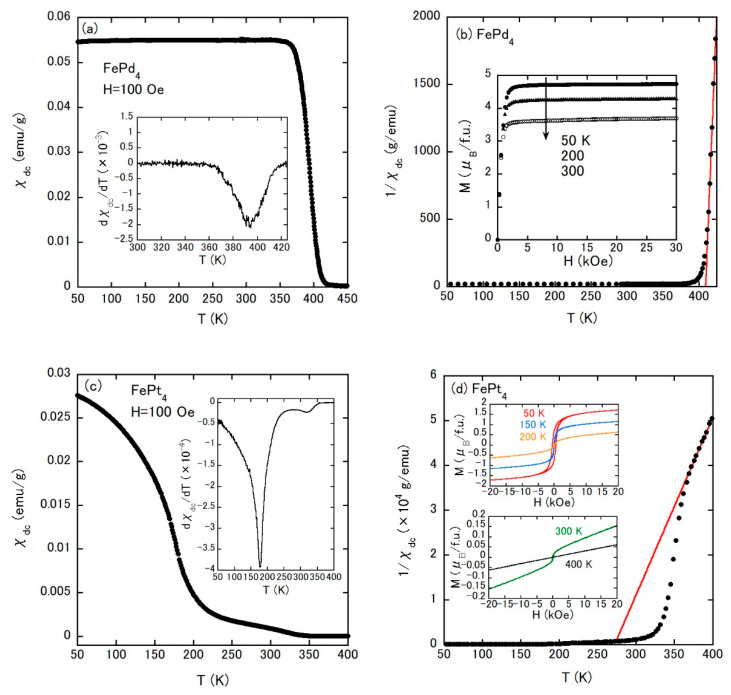
(**a**) Temperature dependence of *χ*_dc_ under an external field of 100 Oe for FePd_4_. The inset is the temperature derivative of *χ*_dc_. (**b**) Temperature dependence of 1/*χ*_dc_ of FePd_4_. The inset shows the isothermal magnetization curves of FePd_4_ at 50, 200, and 300 K. (**c**) Temperature dependence of *χ*_dc_ under an external field of 100 Oe for FePt_4_. The inset is the temperature derivative of *χ*_dc_. (**d**) Temperature dependence of 1/*χ*_dc_ of FePt_4_. The insets are the isothermal magnetization curves of FePt_4_ at 50, 150, 200, 300, and 400 K.

**Figure 5 materials-14-02877-f005:**
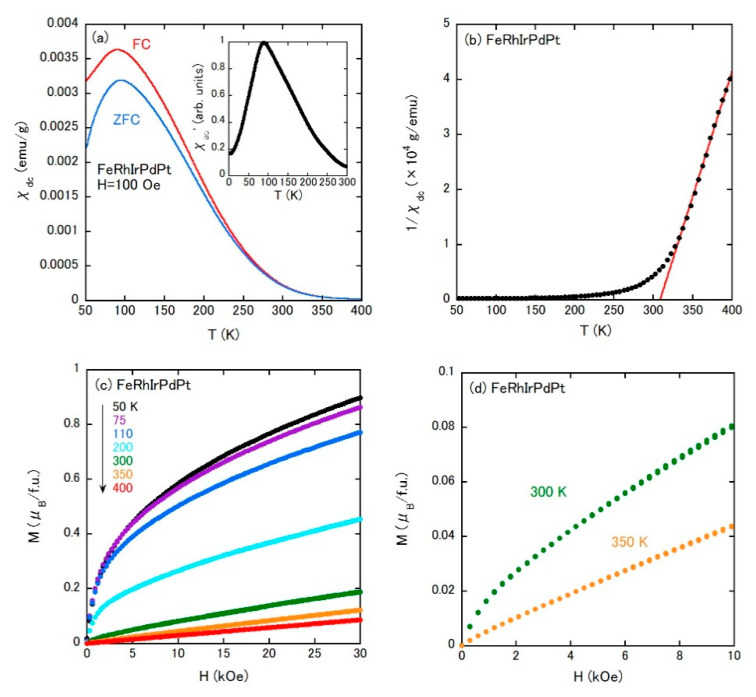
(**a**) Temperature dependence of *χ*_dc_ measured under ZFC and FC conditions for FeRhIrPdPt. The external field is 100 Oe. The inset is the temperature dependence of the real part of the ac susceptibility *χ*_ac_′(*T*) of FeRhIrPdPt under an ac field of 5 Oe at 800 Hz. (**b**) Temperature dependence of 1/*χ*_dc_ measured under FC condition for FeRhIrPdPt. Isothermal magnetization curves of FeRhIrPdPt at 50, 75, 110, 200, 300, 350, and 400 K for (**c**) and at 300 K and 350 K with expanded scale for (**d**).

**Figure 6 materials-14-02877-f006:**
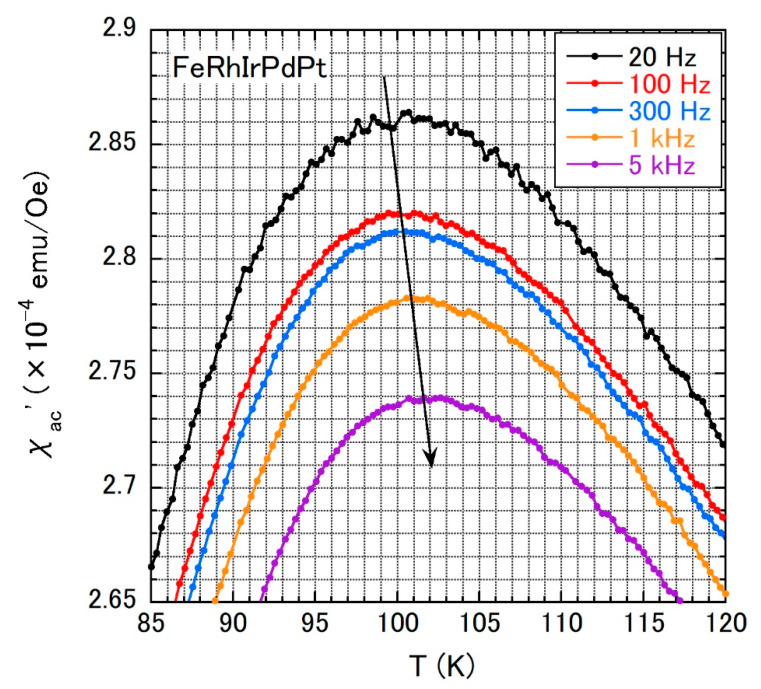
Temperature dependences of *χ*_ac_′ under ac frequencies of 20 Hz, 100 Hz, 300 Hz, 1 kHz, and 5 kHz for FeRhIrPdPt. The arrow is a guide to the eyes.

**Figure 7 materials-14-02877-f007:**
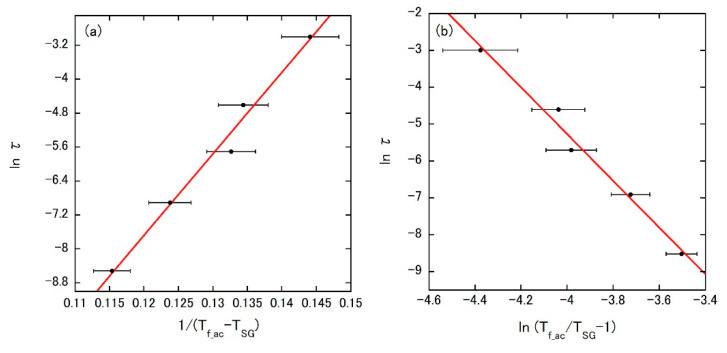
Fitting results of *T*_f_ac_ shift in *χ*_ac_′(*T*) using (**a**) Vogel–Fulcher law and (**b**) critical scaling approach for FeRhIrPdPt, respectively.

**Figure 8 materials-14-02877-f008:**
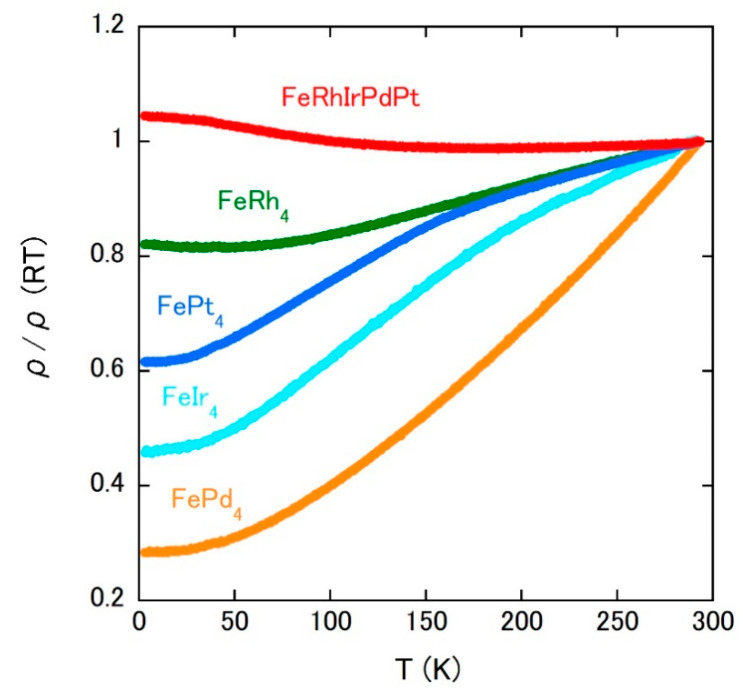
Temperature dependences of electrical resistivity of FeRh_4_, FeIr_4_, FePd_4_, FePt_4_, and FeRhIrPdPt.

**Table 1 materials-14-02877-t001:** Chemical compositions determined by energy-dispersive X-ray spectroscopy and lattice parameters of the prepared samples.

Sample	Composition	Lattice Parameter (Å)
FeRh_4_	Fe_19.2(8)_Rh_80.8(8)_	3.782(1)
FeIr_4_	Fe_18.5(6)_Ir_81.5(6)_	3.807(1)
FePd_4_	Fe_19.3(9)_Pd_80.7(9)_	3.858(1)
FePt_4_	Fe_19.2(6)_Pt_80.8(6)_	3.871(2)
FeIrRhPtPd	main: Fe_15.8(5)_Rh_24.5(5)_Ir_32.6(1)_Pd_10.7(6)_Pt_16.4(6)_ minor: Fe_23.7(7)_Rh_14.0(8)_Ir_5.4(9)_Pd_32.5(9)_Pt_24.4(6)_	3.834(1)

**Table 2 materials-14-02877-t002:** Magnetism, magnetic ordering temperature, effective magnetic moment *μ*_eff_, Weiss temperature *Θ*_CW_, and electrical resistivity ρ at room temperature (RT) of prepared samples. SG, FM, and FM corr. refer to spin-glass, ferromagnetism, and ferromagnetic correlation below 300 K, respectively. *T*_f_ and *T*_C_ are spin-freezing and Curie temperatures, respectively. *T*_f_ is the temperature where field-cooled magnetization under the external field of 100 Oe shows the hump.

Sample	Magnetism	Magnetic Ordering Temperature (K)	*μ*_eff_ (*μ*_B_/Fe)	*Θ*_CW_ (K)	*ρ* (RT) (μΩcm)
FeRh_4_	SG	*T*_f_ = 59	5.74	−700	39.0
FeIr_4_	SG	*T*_f_ = 23, 69	5.71	−890	21.2
FePd_4_	FM	*T*_C_ = 393	5.44	409	59.5
FePt_4_	FM	*T*_C_ = 177, 317	4.11	272	107
FeRhIrPdPt	SG +FM corr.	*T*_f_ = 90	3.40	308	70.8

**Table 3 materials-14-02877-t003:** Fitting parameters obtained by Vogel–Fulcher law and critical scaling approach for FeRhIrPdPt.

Model	*τ*_0_ (s)	*E*_a_ (K)	*T*_SG_ (K)	*zv*
Vogel–Fulcher law	4.8 × 10^−14^	192	93	-
Critical scaling approach	5.1 × 10^−14^	-	90	6.3

**Table 4 materials-14-02877-t004:** Enthalpies of each binary compound. The unit is meV/atom. Data from [38].

	Fe	Rh	Ir	Pd	Pt
Fe	0	−57	−63	−116	−244
Rh	−57	0	−21	37	−24
Ir	−63	−21	0	40	11
Pd	−116	37	40	0	−36
Pt	−244	−24	11	−36	0

## Data Availability

Data sharing is not applicable.

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
