# Peer review of "Magnetic and Transport Properties of New Dual-Phase High-Entropy Alloy FeRhIrPdPt"

_materials, 2021, doi:10.3390/ma14112877_

Round 1

Reviewer 1 Report

Magnetic and transport properties of a new FeRhIrPdPt dual-phase high-entropy alloy is revealed by Baba, et al. The study is interesting and worth publishing. However, major revision is needed.

  1. The XRD result does not exhibit the characteristics of the two-phase high entropy alloy. The authors should provide conclusive evidence for the synthesis of a dual-phase high entropy alloy. For example, TEM images, SAED patterns and elemental mappings are suggested to add into the revision.
  2. Do dual-phase high-entropy alloys have advantages over single-phase high-entropy alloys in magnetic and transport properties? The author needs to have a proper discussion.
  3. What is the preparation process of the dual-phase high entropy alloy in this study? Is there any other treatment towards the sample after arc smelting? What is the solidification condition? What is the sample size? The author should elaborate them in detail.

Author Response

Thank you for your valuable comments. We respectfully respond to your comments.

(1) Comment: The XRD result does not exhibit the characteristics of the two-phase high entropy alloy. The authors should provide conclusive evidence for the synthesis of a dual-phase high entropy alloy. For example, TEM images, SAED patterns and elemental mappings are suggested to add into the revision.

Response: We have added the elemental mappings of FeRhIrPdPt in Fig. 2(f), which supports the dual-phase nature. As the reviewer has suggested, TEM images and SAED patterns are desired. However, we do not possess TEM equipment. So, we have added the sentence “For further elucidation of dual-phase in FeRhIrPdPt, transmission electron microscope images and selected area electron diffraction patterns are desired.” in lines 122-123 of the revised manuscript.

(2) Comment: Do dual-phase high-entropy alloys have advantages over single-phase high-entropy alloys in magnetic and transport properties? The author needs to have a proper discussion.

Response: Following the comment, we have added the sentences “As mentioned in the introduction, dual-phase HEAs would be useful in tailoring magnetic properties via a change of microstructure. In this case, the occurrence of spinodal decomposition is mainly responsible for the change of microstructure. Although the as-cast dual-phase HEA is investigated in this study, it is important to understand the thermal stability of alloy for the issue of microstructure. This is the future task. The other interesting aspect of dual-phase HEAs is a possible multifunctional magnetic material, in which two magnetic phases offer different magnetic functions.” in lines 319-326 of the revised manuscript. We do not want to discuss the comparison between dual-phase HEA and single-phase one at the present stage. We think that dual-phase HEAs possess interesting features (microstructure, multifunction, and so on). We have added the discussion focusing on the features of dual-phase HEAs.  

(3) Comment: What is the preparation process of the dual-phase high entropy alloy in this study? Is there any other treatment towards the sample after arc smelting? What is the solidification condition? What is the sample size? The author should elaborate them in detail.

Response: Following the comment, we have added the detailed explanations “The elemental metals were arc-melted to a button-shaped sample with a mass of 1.5 g and quenched on a water-chilled Cu hearth.” and “In this study, all samples received neither heat treatment nor deformation like rolling.” in Materials and Methods.

Reviewer 2 Report

The article is very well structured, methods are clearly described in the text, design of experiments, analysis and interpretation of the results are really appropriate, findings and results of the authors are very well linked with the research and results find by other researchers in the domain. The article is also very well written in English language and I would like to send my sincere appreciation to the authors for their great effort and valuable work related to the research.

There is one small error in the text in Line 259 - reference in that Line is at Table 4, not Table 1 as it is written. Beside this very small observation, I am strongly support and recommend this article to be published in the Materials journal (MDPI).

Author Response

Thank you for your very positive comments. We respectfully respond to your comment.

(1) Comment: There is one small error in the text in Line 259 - reference in that Line is at Table 4, not Table 1 as it is written.

  Response: We carefully rechecked the sentence. However, Table 1 is correct, and we do not revise.

Reviewer 3 Report

The dynamic and static magnetic properties are described within the framework of molecular field. It should be mentioned in the article. Indeed it is important because when dealing with 3d systems quantum effects are less important than in magnetic systems of lower dimensionality.

Otherwise this paper is clean and deserves publication.

Author Response

Thank you for your valuable comment. We respectfully respond to your comment.

(1) Comment: The dynamic and static magnetic properties are described within the framework of molecular field. It should be mentioned in the article. Indeed it is important because when dealing with 3d systems quantum effects are less important than in magnetic systems of lower dimensionality.

Response: Following the comment, we have added the sentence “Next we will discuss the static and dynamic magnetic properties within the framework of molecular field.” in lines 146-147 of the revised manuscript.

Reviewer 4 Report

Study of two-phase multicomponent alloy based on Fe with fcc-structured noble metals may give new insights into structure and properties of refractory HEAs with potenntial applications. Manusctipt is clear and give complete experimental informatino about preparatinoand properties of new alloys. Magnetic data is quite rare for HEAs and also give new information relevant for the community. I suggest to asign the paper to special issue https://www.mdpi.com/journal/materials/special_issues/composition_structure_properties_high_entropy_alloys

Nevertheless, before considering for the publication several points should be improved.

1. How cell parameters were refined (Table 1)? I suggest to give corresponding fits for HEA showing two-phases. I also suggest to refine relative amounts of both phases in wt.%.

2.I also suggest to give more information about thermal stability of alloys. Were samples annealed after preparation? If yes, how properties and especially phase composition change after annealing?

3.From the text, it is not clear how current phase relates to existing phase diagrmas. I think that short discussion of relevant binary and multicomponent phase diagrams will help to understand current phase.

4. I suggest to reffere to the following articles: Yusenko et al: Materials 13(6) (2020) 1418; Yusenko et al Scripta Materialia 138 (2017) 22–27 

Author Response

Thank you for your valuable comments. We respectfully respond to your comments.

(1) Comment: How cell parameters were refined (Table 1)? I suggest to give corresponding fits for HEA showing two-phases. I also suggest to refine relative amounts of both phases in wt.%.

Response: We used the CellCalc software to refine the parameter. This software requires the crystal system, d-values at XRD peaks, and the corresponding Miller indices. We have added the explanation in lines 111-112 of the revised manuscript. To perform a reliable Rietveld refinement, we need an XRD pattern with a high S/N ratio. Unfortunately, it is difficult to collect an XRD pattern with a high S/N ratio using our X-ray diffractometer. Instead of the Rietveld refinement, we have tried simulating the XRD pattern of FeRhIrPdPt by the superposition of those of main and minor phases under the fixed lattice parameters and weight fractions of two phases. The lattice parameter of each phase was estimated by the composition-weighted average of those of FeRh4, FeIr4, FePd4, and FePt4. The resultant XRD pattern calculated using the PowderCell 2.3 software is shown at the bottom of Fig. 1. The simulated pattern agrees with the experimental one. The explanation was added in lines 128-133 of the revised manuscript. We have added weight fractions of two phases in lines 118-119 and lines 331-332 of the revised manuscript. 

(2) Comment: I also suggest to give more information about thermal stability of alloys. Were samples annealed after preparation? If yes, how properties and especially phase composition change after annealing?

Response: We do not anneal the samples. As the reviewer has suggested, it is important to understand the thermal stability of alloy for further investigation of the dual-phase magnetic HEA. This is the future task. We have added the sentence, explaining the importance of investigations of thermal stability in lines 322-324 of the revised manuscript.

(3) Comment: From the text, it is not clear how current phase relates to existing phase diagrmas. I think that short discussion of relevant binary and multicomponent phase diagrams will help to understand current phase.

Response: Following the comment, we have added the discussion in lines 299-307 of the revised manuscript.

(4) Comment: I suggest to reffere to the following articles: Yusenko et al: Materials 13(6) (2020) 1418; Yusenko et al Scripta Materialia 138 (2017) 22–27

Response: We have added the references (40 and 41).

Round 2

Reviewer 1 Report

The authors have responded to all the reviewers' concerns. In this situation, it can be considered for accepting now.